# Defining the Role of Isoeugenol from *Ocimum tenuiflorum* against Diabetes Mellitus-Linked Alzheimer’s Disease through Network Pharmacology and Computational Methods

**DOI:** 10.3390/molecules27082398

**Published:** 2022-04-07

**Authors:** Reshma Mary Martiz, Shashank M. Patil, Mohammed Abdulaziz, Ahmed Babalghith, Mahmoud Al-Areefi, Mohammed Al-Ghorbani, Jayanthi Mallappa Kumar, Ashwini Prasad, Nagendra Prasad Mysore Nagalingaswamy, Ramith Ramu

**Affiliations:** 1Department of Biotechnology and Bioinformatics, JSS Academy of Higher Education and Research, Mysuru 570015, Karnataka, India; reshmamarymartiz@gmail.com (R.M.M.); shashankmpatil@jssuni.edu.in (S.M.P.); 2Department of Microbiology, JSS Academy of Higher Education and Research, Mysuru 570015, Karnataka, India; ashwinip@jssuni.edu.in; 3Department of Environmental Health, Faculty of Public Health and Health Informatics, Umm Al-Qura University, Makkah 21955, Saudi Arabia; maabdulaziz@uqu.edu.sa; 4Department of Medical Genetics, Faculty of Medicine, Umm Al-Qura University, Makkah 21955, Saudi Arabia; aobabalghith@uqu.edu.sa; 5Department of Health Information Management & Technology, Faculty of Public Health and Health Informatics, Umm Al-Qura University, Makkah 21955, Saudi Arabia; maareefi@uqu.edu.sa; 6Department of Chemistry, College of Science and Arts, Ulla, Taibah University, Madina 41477, Saudi Arabia; malghorbani@yahoo.com; 7Department of Chemistry, College of Education, Thamar University, Thamar 425897, Yemen; 8Department of Pharmacology, JSS Medical College, JSS Academy of Higher Education and Research, Mysuru 570015, Karnataka, India; mkjayanthi@jssuni.edu.in; 9Department of Biotechnology, JSS Science and Technology University, JSS Technical Institutional Campus, Mysore 570006, Karnataka, India; npmicro8@yahoo.com

**Keywords:** *Ocimum tenuiflorum*, diabetes mellitus, Alzheimer’s disease, GAPDH, AGE-RAGE, isoeugenol, network pharmacology, druglikeness and pharmacokinetics, molecular docking simulation, GO analysis, molecular dynamics simulation, binding free energy analysis

## Abstract

The present study involves the integrated network pharmacology and phytoinformatics-based investigation of phytocompounds from *Ocimum tenuiflorum* against diabetes mellitus-linked Alzheimer’s disease. It aims to investigate the mechanism of the *Ocimum tenuiflorum* phytocompounds in the amelioration of diabetes mellitus-linked Alzheimer’s disease through network pharmacology, druglikeness and pharmacokinetics, molecular docking simulations, GO analysis, molecular dynamics simulations, and binding free energy analyses. A total of 14 predicted genes of the 26 orally bioactive compounds were identified. Among these 14 genes, GAPDH and AKT1 were the most significant. The network analysis revealed the AGE-RAGE signaling pathway to be a prominent pathway linked to GAPDH with 50.53% probability. Upon the molecular docking simulation with GAPDH, isoeugenol was found to possess the most significant binding affinity (−6.0 kcal/mol). The molecular dynamics simulation and binding free energy calculation results also predicted that isoeugenol forms a stable protein–ligand complex with GAPDH, where the phytocompound is predicted to chiefly use van der Waal’s binding energy (−159.277 kj/mol). On the basis of these results, it can be concluded that isoeugenol from *Ocimum tenuiflorum* could be taken for further in vitro and in vivo analysis, targeting GAPDH inhibition for the amelioration of diabetes mellitus-linked Alzheimer’s disease.

## 1. Introduction

There have been several insights into the exploration of the biomolecular link between type 2 diabetes mellitus (T2D) and Alzheimer’s disease (AD) in the last two decades [1]. However, there is no specific molecular pathway or target that has been identified for the development of AD in diabetic patients. Epidemiological details suggest a strong relationship between high blood glucose levels and the degree of cognitive dysfunction in diabetic patients [2,3]. A few studies have collectively termed these conditions as type 3 diabetes [4,5].

For the treatment of T2D-linked AD, there are no specific therapeutic applications available. However, biguanides, sulfonylureas, glitazones, and DPP-4 inhibitors are known to reduce the symptoms of AD. However, most of these drugs have been reported with adverse effects, such as weight loss, lactic acidosis, ketoacidosis, anemia, bone fractures, and gastrointestinal and cardiovascular complications. Therefore, it is worth testing phytoconstituents with antidiabetic potential, which are less toxic compared to synthetic drugs [6,7]. 

*Ocimum sanctum*, commonly known as *tulsi*, is a fragrant shrub of the basil family *Lamiaceae* (*Tribeocimeae*) that is native to the eastern globe tropics, and which is said to have originated in north-central India. *Tulsi* is supposed to help with disease prevention, general health, wellbeing, and longevity, as well as dealing with the demands of daily life [8]. Specifically, it is also said to have an ameliorating effect on several health maladies [9,10,11]. Several in vitro, animal, and human experiments have proven the pharmacological benefits of tulsi [10,11,12]. Therefore, *O. sanctum* serves as an important source of phytoconstituents that can be utilized for pharmacotherapeutic applications.

With the advancements of computer-aided drug design technologies, several computational investigations have identified phytoconstituents as an effective replacement for the available therapeutics [13]. Additionally, network pharmacology has been employed to study the biological network of drug candidates, and to construct a poly-target drug molecule to optimize medication efficacy [14,15,16]. Hence, we aim to investigate the anti-diabetic as well anti-Alzheimer’s potential of *O. sanctum* phytochemicals through network pharmacology and other computational methods.

## 2. Materials and Methods

### 2.1. Construction of Phytocompound Library

The phytocompounds of *Ocimum tenuiflorum* were searched for through IMPPAT (Indian Medicinal Plants, Phytochemistry and Therapeutics) database, (https://cb.imsc.res.in/imppat/basicsearchauth) (Accessed on 3 January 2022) [17], which contains more than 1700 Indian medicinal plants with 1100 therapeutic uses. The phytocompounds identified were retrieved from the NCBI PubChem database (https://pubchem.ncbi.nlm.nih.gov/) (Accessed on 3 January 2022). On the basis of their ADMET properties, the active compounds were screened for multiple parameters, such as oral bioavailability (OB) ≥ 30%, blood–brain barrier (BBB), half-life (HL) < 3 h, Lipinski’s rule of five, Caco-2 cells, drug-induced liver injury (DILI), clearance (CL) > 15 mL, molecular weight (MW), hydrogen bond donors (HbD), hydrogen bond acceptors (HbA), topological polar surface area (TPSA), and pan assay interference compounds (PAINS).

### 2.2. Prediction of Target Genes

The genes associated with type 2 diabetes and AD were obtained using DisGeNET (https://www.disgenet.org/search) (Accessed on 5 January 2022) and the GeneCards database (https://www.genecards.org/) (Accessed on 3 January 2022). The target genes were searched for using keywords “T2D” and “Alzheimer’s”, using “*Homo sapiens*” as a filter for both.

### 2.3. Construction of PPI Network

The protein interaction network was built for the common genes taken from both the diseases, to understand their functional associations and the extension of interactions, using the STRING online tool (https://string-db.org) (Accessed on 7 January 2022). The STRING database was used to remove dissociation targets from the target genes, with a confidence level of 0.09 and an organism filter set to Homo sapiens. Finally, the interaction network was constructed.

### 2.4. C-T Network and Enrichment Analysis

The network obtained from STRING was imported to Cytoscape 3.8.2 [18]. By using the CytoCluster plugin, the gene cluster was obtained to find the highly significant genes. Later, the core targets were selected using the most important parameters on the basis of the topological characteristics of the constructed networks, such as Degree Centrality (DC), which is used to evaluates the number of nodes associated with a node, Closeness Centrality (CC), which refers to the sum from one point to another, and Betweenness Centrality (BC), which is used to evaluate the shortest path in the network [19] using the CytoNCA plugin. Finally, the selected genes were again screened using CytoHubba plugin using the Maximum Clique Centrality (MCC) scoring method to generate the critical subnetwork. The biological interpretation of core target genes was performed using the ClueGO plugin, which integrates the GO and KEGG pathways and creates a functionally organized network, further using CluePedia to identify the potential biomarkers that are associated with the pathway, as well as the cellular location of the core genes analyzed [20]. The filter for analysis was set to *Homo sapiens* with a *p* value less than 0.05, and the Bonferroni method was considered. Finally, the C-T network was built for the core targets and was analyzed using Cytoscape 3.8.2.

### 2.5. Molecular Docking Simulation

The 3D crystal structure of the target protein GAPDH (PDB ID: 1U8F) was downloaded from the RCSB Protein Data Bank (https://www.rcsb.org/) (Accessed on 10 January 2022). The molecular docking protocol was validated by redocking the co-crystallized ligand with the target protein and evaluating the RMSD between the bound and re-docked ligand. The molecular docking simulation was carried out according to the previous study conducted by Patil et al. (2021) [21]. For the molecular docking simulation, AutoDock Tools 1.5.6 was used to prepare the protein structures. AutoDock is a suite of automated docking tools. It is designed to predict how small molecules, such as substrates or drug candidates, bind to a receptor of a known 3D structure [22]. To purify the protein structures, water and heteroatoms were removed. Polar hydrogens, on the other hand, were added to stabilize the same. The energy of the protein structures was reduced by using Kollmann-united charges and Gasteiger charges. After energy minimization, all atoms were assigned an AutoDock 4 atom type prior to obtaining the prepared protein structure in the PDBQT format for molecular docking simulations. The PDBQT format saves the protein’s atomic coordinates, partial charges, and AutoDock 4 atom type [23]. Further, OpenBabel 2.3.1 was used to transform the 3D SDF structures into PDBQT format during ligand preparation [24]. For ligand preparation, the structures were imported into AutoDock Tools 1.5.6. To reduce the energy of the ligands, Kollman-united and Gasteiger charges were added. As indicated by AutoDock Tools 1.5.6, the number of torsions was left at the default [25].

Binding site prediction for the target proteins was completed using a literature analysis. The binding residues present in the binding pocket were identified using the literature available for GADPH [26]. Using AutoDock Tools 1.5.6, the binding pockets of the respective proteins were set in the grid boxes. The grid box measuring 11.20 Å × 11.20 Å × 11.20 Å was coordinated at x = 12.128023, y = 24.467386, and z = 29.015455. Virtual screening of the phytocompounds was completed with a command line-based software known as AutoDock Vina 1.1.2. For the perturbation and local optimization of ligands into the target site, it employs the Broyden–Fletcher–Goldfarb–Shanno (BFGS) algorithm, which analyses the scoring function of each ligand conformation [27]. Because of the number of torsions generated during ligand production, ligands were assumed to be flexible throughout the docking simulation, while the proteins were believed to be rigid. However, 10 degrees of freedom were permitted for ligand molecules. The first binding poses with zero root-mean-square deviation (RMSD) of atomic positions are deemed to be highly valid out of ten binding poses generated [28]. They also possess the strongest binding affinity of all the positions, implying that the binding is stronger. Biovia Discovery Studio Visualizer 2021, an open-source visualizing GUI software, was used to visualize the results. The extent of ligand interaction was determined using binding affinity, total number of bonds, and respective hydrogen bonds [29].

### 2.6. Molecular Dynamics Simulation

The docked conformations of GAPDH protein and respective ligands (isoeugenol and myrene) with the lowest negative binding affinity were chosen for the molecular dynamics simulation, which was carried out according to the previous study conducted by Patil et al. (2021) [30]. Myrene was chosen as the negative control, since the molecule was predicted to have the highest positive binding affinity (inferior binding) with GAPDH. For the simulation, the GROMACS-2018.1 biomolecular software suite was utilized [31]. All of the protein–ligand complexes were assigned using the CHARMM36 force field, and the ligand topology was acquired using the CGenFF server [32]. The GROMACS pdb2gmx module was used to add hydrogen atoms to the heavy atoms present. The steepest descent technique was then utilized to achieve 5000 vacuum minimization steps. The protein–ligand complexes were placed in a box with a 10 Å perimeter. The solvent was included in the water model TIP3P. The entire system was neutralized by adding the proper amount of Na^+^ and Cl^−^ counter ions. A total of 3 simulations were run for 100 ns, including a GAPDH protein backbone atom (5051 residues), GAPDH protein-isoeugenol complex (5065 residues), and GAPDH protein–myrene complex (5062 residues). The energy of the generated systems was decreased using the steepest descent and conjugate gradient techniques. The NVT ensemble was then followed by an NPT ensemble, which was followed by a brief (1000 ps) equilibration (1000 ps). All simulations took 100 ns at 310 K temperature and 1 bar pressure. A trajectory analysis consisting of root-mean-square deviation (RMSD), root-mean-square fluctuation (RMSF), radius of gyration (Rg), and solvent accessible surface area (SASA), and ligand hydrogen bond parameters was performed using the XMGRACE software, with the results displayed in graphical form [33,34].

### 2.7. Binding Free Energy Calculations

The determination of a protein–ligand complex’s binding free energy is another application of molecular dynamics simulations and thermodynamics for determining the extent of ligand binding with protein. The binding free energy calculations for all the protein–ligand complexes were evaluated according to the previous study by Patil et al. (2021) [30]. The molecular mechanics/Poisson–Boltzmann surface area (MM-PBSA) technique was used in this work. It is another application of molecular dynamics simulations and thermodynamics for determining the extent of ligand binding with protein [35,36]. The g_mmpbsa program with MmPbSaStat.py script, which utilizes the GROMACS 2018.1 trajectories as input, was used to determine the binding free energy for each ligand–protein combination. In the g_mmpbsa program, three components are used to calculate the binding free energy: molecular mechanical energy, polar and apolar solvation energies, and molecular mechanical energy. The calculation is performed using MD trajectories of the last 50 ns, which were considered to compute ΔG with dt 1000 frames. It is evaluated using molecular mechanical energy and polar and apolar solvation energies. The equation (I) and (II) that are used to calculate the free binding energy are given below [21,30].
ΔG_Binding_ = G_Complex_ − (G_Protein_ + G_Ligand_)(1)
ΔG = ΔE_MM_ + ΔG_Solvation_ − TΔS = ΔE_(Bonded + non-bonded)_ + ΔG_(Polar + non-polar)_ − TΔS(2)

G_Binding_: binding free energy, G_Complex_: total free energy of the protein–ligand complex, G_Protein_ and G_Ligand_: total free energies of the isolated protein and ligand in solvent, respectively, ΔG: standard free energy, ΔE_MM_: average molecular mechanics potential energy in vacuum, G_Solvation_: solvation energy, ΔE: total energy of bonded as well as non-bonded interactions, ΔS: change in entropy of the system upon ligand binding, and T: temperature in Kelvin.

## 3. Results

### 3.1. Target Prediction and Active Compound Library

A total of 61 compounds of *Ocimum tenuiflorum* were retrieved from the PubChem database. On the basis of the ADMET (Table 1) screening, an active compound library was built, and a total of 26 compounds passed the screening criteria. Further, the target genes were fished for using the DisGeNET database, and a total of 6531 were considered, where 3134 belong to T2D and 3397 belong to AD, respectively. However, only 1381 genes were found to be common in both the diseases (Figure 1).

### 3.2. Construction of PPI Network

The interaction network was constructed for the target proteins. Out of 1381 common genes, the network was constructed for 1287 targets which were specific to Homo sapiens. The constructed network contained 1287 nodes and 5685 edges with the 8.83 as the average node degree and 0.564 as the coefficient. The pictorial representation of the diseases interaction network is given in Figure 2.

### 3.3. Network Analysis

The disease gene interaction network obtained from STRING was imported to Cytoscape 3.8.2 for further analysis. The interaction network that was imported to Cytoscape contained 1025 nodes and 5685 edges. By using the CytoCluster plugin, the gene cluster was obtained to find the highly significant genes. A total of 157 clusters were obtained, out of which 19 had significant *p* values of less than 0.05, as depicted in Figure 3. Further, using the CytoNCA plugin, the top 15 genes were selected (Table 2), and again the selected genes were screened using the CytoHubba plugin to obtain the critical genes. After all three analyses, the top 14 genes were selected for the construction of a C-T network. The compounds were selected on the basis of the above ADMET screening.

After the discovery of core target and active compounds, the C-T network was constructed, as shown in Figure 4. The interaction network contains 35 nodes and 196 edges, which were analyzed using Cytoscape. The analysis shows that the phytocompounds of *Ocimum tenuiflorum* are connected to the target genes. Among these 14 genes, GAPDH and AKT1 were the most significant, according to the analysis.

### 3.4. Gene Ontology and Pathway Enrichment Analysis

The top 14 core genes were then analyzed using ClueGo and CluePedia for the GO functional annotation and KEGG pathway analysis. On the basis of the analysis result, a total of 148 pathways were obtained, which are divided into 17 functional groups that represents the pathways with GO terms as depicted in the pie chart (Figure 5). Figure 6 shows the graphical representation of the cellular location of the potential marker using the interaction score from the GO term. The outcome of GO and pathway enrichment reveals the AGE-RAGE signaling pathway to be the most significant target pathway with 50.53% probability in comparison with the other pathways analyzed (Figure 5).

### 3.5. Molecular Docking Study

Following the network analysis, the most significant gene glyceraldehyde 3-phosphate dehydrogenase (GAPDH) was considered for the docking study. The structural details of the selected protein molecule (PDB ID: 1U8F) have been depicted in Appendix A. The results from the molecular docking protocol reveal that the protocol was suitable for the screening of the phytocompounds. The RMSD between the bound and re-docked ligand was found to be 2.10 Å (Appendix A). We then selected the active compounds for the docking study on the basis of the C-T network and ADMET screening, and a total of 11 compounds were considered for the study. The virtual screening result is given in Table 3. The screening result showed that the GAPDH-bound isoeugenol complex showed a better binding affinity of −6.0 kcal/mol with the highest non-bonded interaction than other compounds. However, myrcene was found to have the lowest binding affinity (−3.6 kcal/mol) and was considered to be a negative control for further analyses. Both isoeugenol and myrcene were bound to the same binding site, where the co-crystallized NAD^+^ was bound. Isoeugenol formed a total of eight non-bonding interactions, with three of them being hydrogen bonds. It bound to ARG 13 (3.20 Å), GLY 12 (3.41 Å), and SER 98 (2.03 Å) with hydrogen bonds. It also interacted with the protein using hydrophobic pi-alkyl bonds with ARG 13 (4.27 Å and 5.10 Å), ILE 14 (5.24 Å and 4.98 Å), and an alkyl bond with ALA 183 (3.90 Å). However, myrcene was found to have only two non-bonding interactions, both being alkyl bonds. It bound only with ILE 14 (5.14 Å and 4.51 Å) (Figure 7). A protein–ligand interaction fingerprint (PLIF) was conducted to interpret the docking results (Appendix A).

### 3.6. Molecular Dynamics Simulation

A molecular dynamics simulation was used to validate the docking investigation and determine the degree of stability of the docked complex along with the target protein. Graphs for RMSD, RMSF, Rg, SASA, and H-bond are represented in Figure 8 to analyze the trajectories.

The root-mean-square deviation (RMSD) graph represents the protein–ligand complex’s stability throughout the course of a 100 ns simulation. By examining the plot, it can be said that the complex and target protein show a similar pattern of stability. The complex is within the range of 0.2–0.5 nm and the protein backbone was found within 0.2–0.35, and both were found to be stable after 80 ns. However, the protein–myrcene complex was found to have an RMSD value of 0.6 nm, and the plot was not concordant with that of the protein backbone atoms. In the RMSF analysis, both the isoeugenol complex and the protein backbone atoms were on par, with an almost similar fluctuation pattern. In the protein–myrcene complex, a higher number of fluctuations was observed in the loop regions (200 residues). Fluctuations were also found between 50 and 100 residues in 200–250 regions. Further, Rg and SASA plots were analyzed to show the structural compactness of the structure formed. The Rg plot analysis shows that the protein and the complex were found to be ranging between 2.0 and 2.1 nm, whereas the SASA value was found to be on par, with a similar pattern. The same trend of incompatible plot was predicted for protein–myrcene complex in Rg analysis, which was observed in the case of RMSD and RMSF. It was equilibrated at 2.2–2.3 nm. In the SASA plots, the value of protein–myrcene complex was found to be ranging between 130 and 135 nm^2^, which made the plot incompatible with that of the protein backbone atoms. Finally, the H-bond was assessed to determine the structural re-agreement, and it can be seen that the complex may have undergone structural alterations. In ligand hydrogen bond formation, isoeugenol formed more hydrogen bonds (six) in comparison with myrcene (four).

### 3.7. Binding Free Energy Calculations

To determine the energy generated during the MD simulation, the free binding energy calculation was performed for both protein–ligand complexes. From the analysis, it can be seen that, aside from polar solvation energy and electrostatic energy, other energies contribute negatively to the complex development in the case of both the complexes. From the results, it is clear that the GAPDH–isoeugenol complex is more stable than the GAPDH–myrcene complex. Apart from the experimental values, the standard deviations also indicate the deviation of the GAPDH–myrcene complex from stabilization. Table 4 gives the details of the energy formed, along with their values.

## 4. Discussion

Over the past few decades, there have been a substantial number of studies conducted on the relation between type 2 diabetes (T2D) and AD (AD), as both T2D and AD are age-related conditions [37,38,39]. According to a study that has been recently conducted [40], it is known that there is about a 50–150% chances of an increase in cases of AD with people suffering from T2D compared with the normal population. T2D affects, directly or indirectly, various metabolic, inflammatory, signaling, and hyperinsulinemia functions, which can also be a related risk factor leading to AD. One of the most common features of T2D is impairments in insulin actions and signaling, which leads to hyperglycemia and hyperinsulinemia [40]. The most common factor linked to T2D and AD is the role of insulin resistance [3,40].

*Ocimum tenuiflorum*, which is commonly called holy basil or tulsi, belongs to the Lamiaceae family. Over the past few decades, many researchers have studied the pharmacological properties of *tulsi*, and it has been used in Ayurveda up until now. In accordance with [41,42,43], *tulsi* has exhibited a beneficial effect on both diabetes and AD. With this in consideration, we used phytocompounds from tulsi to study the effect of potential anti-diabetic and anti-Alzheimer’s drugs.

To understand the relationship between T2D and AD, the alternative herbal medicine network pharmacology approach was used. The core target and the active compounds were identified to understand the disease network and their pathway from the network perspective. The PPI network for the common genes from both T2D and AD was constructed using the STRING database, which was later imported to Cytoscape for analysis. The main focus of this objective was to find the relationship through network construction and to find the most significant genes and their pathway using GO terms.

The constructed PPI contained a total of 19 significant clusters that had *p* values of less than 0.05 (*p* < 0.05). Further, using CytoNCA and CytoCluster, the most significant GAPDH gene was considered for a docking and dynamic study to understand the interaction of active compounds on GAPDH as target. In an in vivo study, GAPDH was reported to play a significant role in the development of diabetic retinopathy by elevating retinal mitochondrial superoxide levels. It is also reported to elevate the AGE-RAGE pathway, protein kinase C, and the hexosamine pathways, which are considered to be the hallmarks of T2D [44,45,46]. However, in the presence of αβ-42, the nitrosylation of GAPDH can increase tau acetylation, causing a disruption in the microtubule association process, and the amount of nitrosylated GAPDH is elevated in AD brains. These studies indicate that GAPDH is linked with both T2D and AD and could be used as a potential target. Thus, the inhibition of GAPDH would be an essential step to reduce the pathogenesis of diabetes-linked AD [47].

We screened the core 14 genes for the pathway analysis with GO term using ClueGo and Cluepedia plugins. The core 14 genes are correlated with multiple biological processes and pathways, which includes the AGE-RAGE signaling pathway, the interleukin-4 and interleukin-13 signaling pathways, the photodynamic therapy-induced NF-kB survival signaling pathway, the negative regulation of the lipid catabolic process, and pancreatic cancer, which are the top interacting pathways. Our study shows that the AGE-RAGE signaling pathway may play a key role in drug development for both T2D and AD.

In both hyperglycemic and calcification settings, AGE/RAGE signaling promotes both cellular and systemic responses to increase bone matrix proteins via the PKC, p38 MAPK, fetuin-A, TGF-, NFB, and ERK1/2 signaling pathways. Through the activation of Nox-1 and the decreased expression of SOD-1, AGE/RAGE signaling has been demonstrated to increase oxidative stress and promote diabetes-related vascular calcification. Increased oxidative stress caused by AGE/RAGE signaling in diabetes-related vascular calcification was also linked to the phenotypic transformation of VSMCs to osteoblast-like cells in AGE-induced calcification. Researchers discovered that pharmacological drugs and antioxidants reduced calcium deposition in the diabetic vascular calcification caused by AGEs [48]. However, GAPDH is linked with the activation of the AGE-RAGE pathway, which could be the initiating point of diabetes-linked AD pathogenesis [44,45].

The most significant gene, GAPDH, was selected as a target of an interaction study after C-T network construction. Before going for the molecular docking simulation, ADMET profiling of the retrieved compounds was performed. Compounds that obey the desired ADMET properties were selected for further analysis. The Lipinski rule of five was considered as an important parameter for this analysis, as it deals with the druglikeness parameters. Although most of the compounds accepted the rule, other pharmacokinetic rules, such as oral bioavailability, the blood–brain barrier, drug half-life, and drug-induced liver injury, were considered for the screening of the compounds. This was conducted because natural products are often cited as an exception to Lipinski’s rules. Phytochemicals are considered to be the product of nature’s ‘evolution’, which enhances the functionality of the compounds in specific metabolic pathways [49]. When it comes to making physiologically active molecules with high molecular weights and a large number of rotatable bonds, nature has learned to maintain low hydrophobicity and intermolecular H-bond donating potential. Natural products are also more likely than pure synthetic compounds to resemble biosynthetic intermediates or endogenous metabolites, allowing them to benefit from active transport systems. Surprisingly, the natural product leads in the Lipinski universe all delivered an oral medication with a 50% success rate [50].

Only selected compounds from the ADMET screening were docked with the target protein. From the docking study, we see that isoeugenol had a better binding affinity of 6.0 kcal/mol with the highest non-bonded interaction and hydrogen bonds of seven and two. Where the hydrogen bond formed is within the active site of the structure. The binding of the isoeugenol to the hydrophobic site, which is supposed to be occupied by nicotinamide adenine dinucleotide phosphate (NADPH), suggests that isoeugenol inhibits the enzyme activity. Moreover, other known details of the hydrophobic site also mention the same [51]. Among the docked compounds, myrcene was predicted with the lowest (more positive) binding affinity (−3.6 kcal/mol). It had only two non-bonding interactions with zero hydrogen bonds. Therefore, it was further considered as a negative control for MD simulation. Although geranyl acetate was predicted with −5.9 kcal/mol of binding affinity, which is close to that of the isoeugenol’s, the compound was not considered to be a computational hit. This was because of the comparatively fewer non-bonding interactions (six) and hydrogen bonds (one) than isoeugenol. In the PLIF analysis, binding residues interacting with isoeugenol and myrcene were used, for which residues interacting with NAD^+^ were used as a reference (Appendix A). However, ALA 183 was not considered for PLIF, since it does not interact with NAD^+^. A study conducted on GAPDH inhibition by 3-bromopyruvate and its derivatives showed that molecular docking can be performed for this protein [52]. However, the study failed to provide sufficient proof for the stability of the experimental compounds through MD simulation and binding free energy calculations.

Both isoeugenol and myrcene were found to obey Lipinski’s rule of five. Both the compounds passed the oral bioavailability parameter. Isoeugenol was predicted to have negative drug-induced liver injury (DILI), yet myrcene was found to have moderate DILI. Therefore, isoeugenol could be a better drug candidate than myrcene in terms of cytotoxicity. Additionally, the CYP450 metabolism of isoeugenol generates a corresponding epoxide, which could act as a potential inhibitor of GAPDH. This is due to GAPDH’s increased proclivity for reacting with epoxide as a result of its contact with the phosphate moiety, which causes the enzyme to change its conformation [53].

Further, to know the stability of the docked complex, an MD simulation was performed. From the MD result, we can see that the complex formed is stable, and a structural re-arrangement may have taken place with the docked structure. In the MD result analysis, isoeugenol-bound GAPDH was found to be more stable in comparison with myrcene-bound GAPDH. The graphical analysis of the MD trajectories, including RMSD, RMSF, Rg, SASA, and ligand hydrogen bonds, depict the stability of isoeugenol inside the binding pocket over myrcene. The binding free binding energy was calculated, as it is known that the greater the negative value, the higher the stability. The calculated result had the free binding energy of −143.458 kj/mol; thus, we can conclude that the docked structure is stable. However, the GAPDH–isoeugenol complex is chiefly formed by the van der Waal’s energy, which is −159.277 kj/mol. However, the binding free energy values of the GAPDH–myrcene complex were found to be inferior in comparison with the GAPDH–isoeugenol complex. Although both the complexes were formed majorly using van der Waal’s energy, the GAPDH–isoeugenol complex was predicted to have more stability. Results obtained from both molecular dynamics simulation and binding free energy calculations support the molecular docking simulation results. The present investigation is the first ever to be conducted on GAPDH inhibition using plant-based compounds through network pharmacology and computational studies. Until now, there have been no studies on GAPDH inhibition through an in-silico approach using phytochemicals. With these findings, we report the pharmacological significance of isoeugenol from *Ocimum tenuiflorum against diabetes-linked AD targeting GAPDH.*

## 5. Conclusions

A phytochemical intervention into T2D has already been proven to ameliorate the lifestyle-based disorder. However, diabetes-linked AD and other diabetic complications have not been focused on so far. Therefore, in this study, we report the virtual screening of isoeugenol from *Ocimum tenuiflorum* as an inhibitor of GAPDH, a common target for both T2D and AD. The study systematically identifies GAPDH as a promising target through network pharmacology tools. However, the phytocompounds have also been screened through a druglikeness and pharmacokinetic approach. Moreover, the compound target network also identifies isoeugenol as the best inhibitor of GAPDH. This hypothesis is supported by the computational tools, including a molecular docking simulation, a molecular dynamics simulation, and binding free energy calculations, where the binding interactions and stability of the isoeugenol with GAPDH have been proven to be satisfactory. With these results, we can conclude that isoeugenol could be the lead potential inhibitor of GAPDH. In future, isoeugenol could be used for in vitro and in vivo studies against GAPDH, targeting T2D-linked AD.

## Figures and Tables

**Figure 1 molecules-27-02398-f001:**
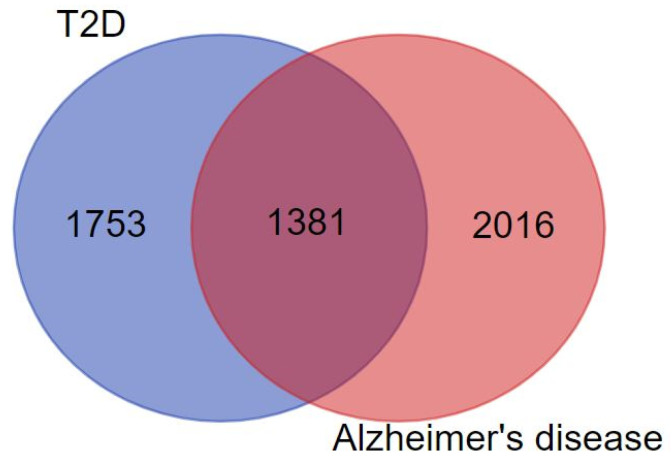
Venn diagram representing the interaction of T2D with AD.

**Figure 2 molecules-27-02398-f002:**
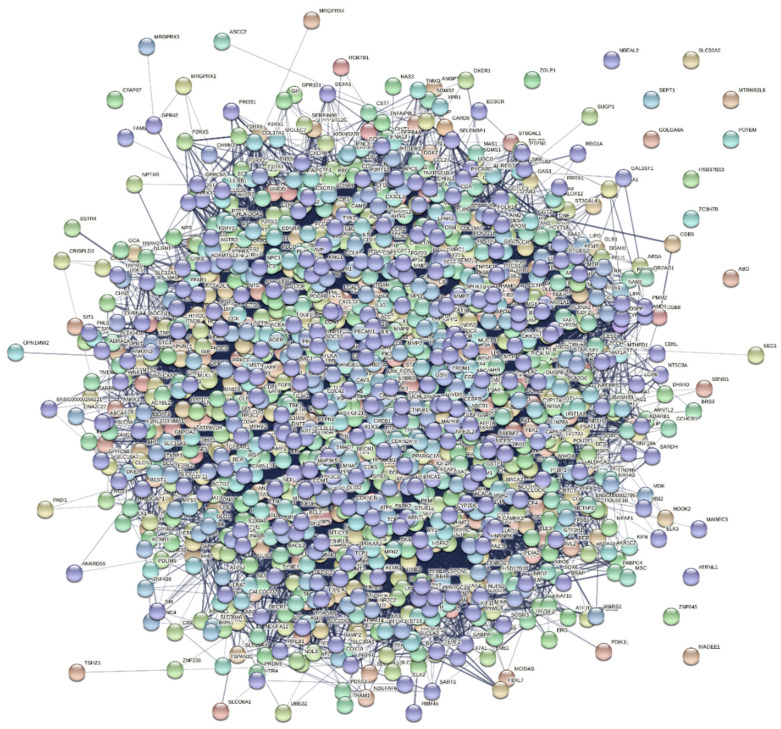
The interaction network of common genes built using STRING for T2D and AD.

**Figure 3 molecules-27-02398-f003:**
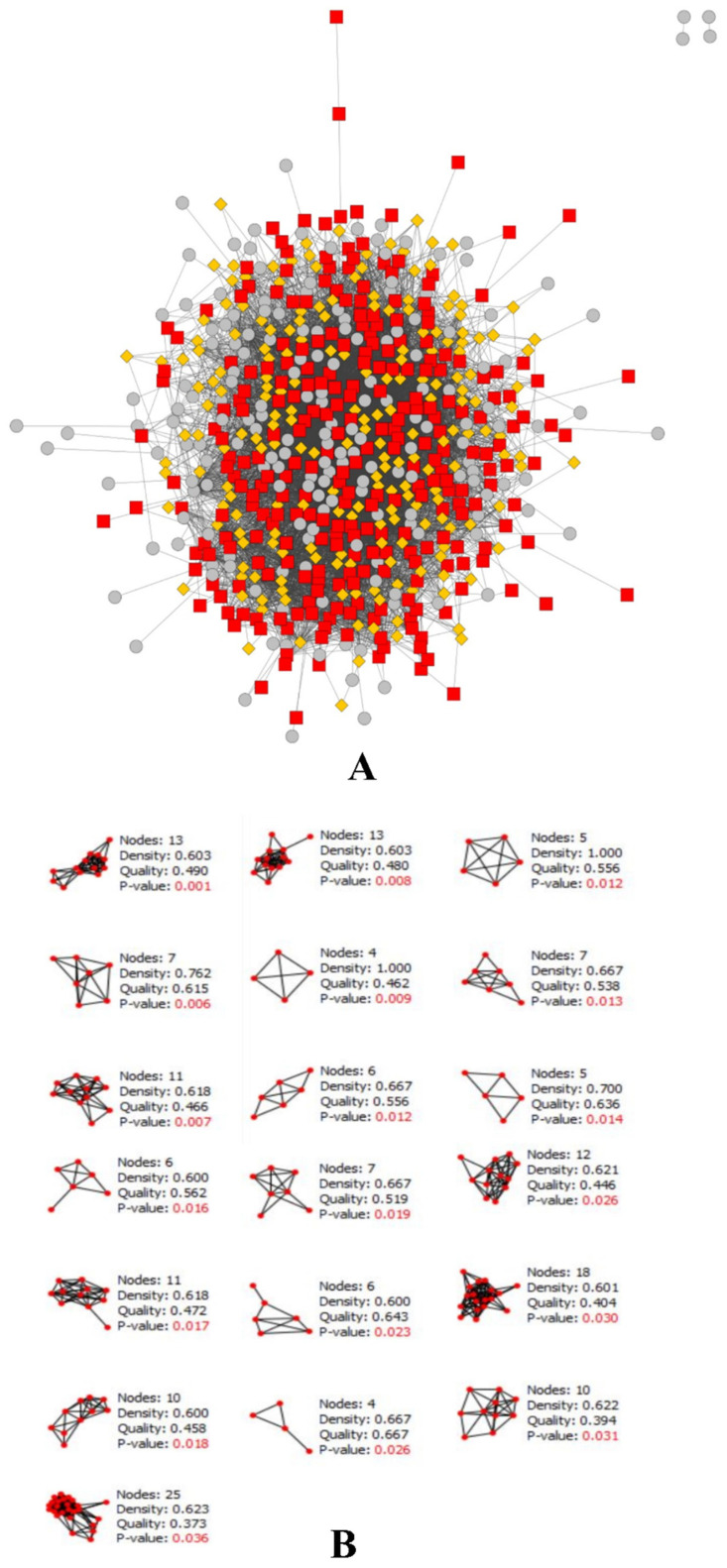
The (**A**) CytoCluster result, with highly significant clusters shown in red; (**B**) the significant clusters with *p*-values of less than 0.05 obtained after analysis.

**Figure 4 molecules-27-02398-f004:**
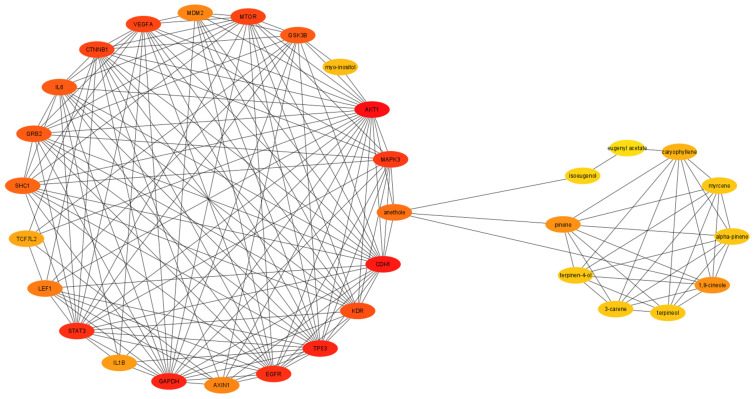
Compound target network obtained using Cytoscape (significance based on color intensity).

**Figure 5 molecules-27-02398-f005:**
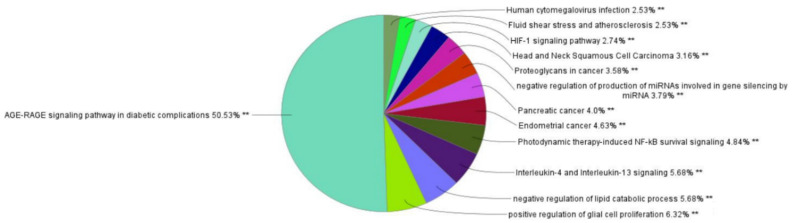
The pie chart result of GO term analysis with interaction pathway obtained using ClueGo.

**Figure 6 molecules-27-02398-f006:**
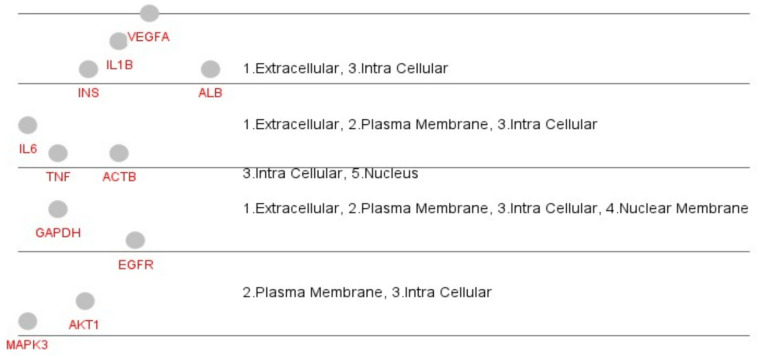
The cellular location of the potential marker based on CluePedia analysis.

**Figure 7 molecules-27-02398-f007:**
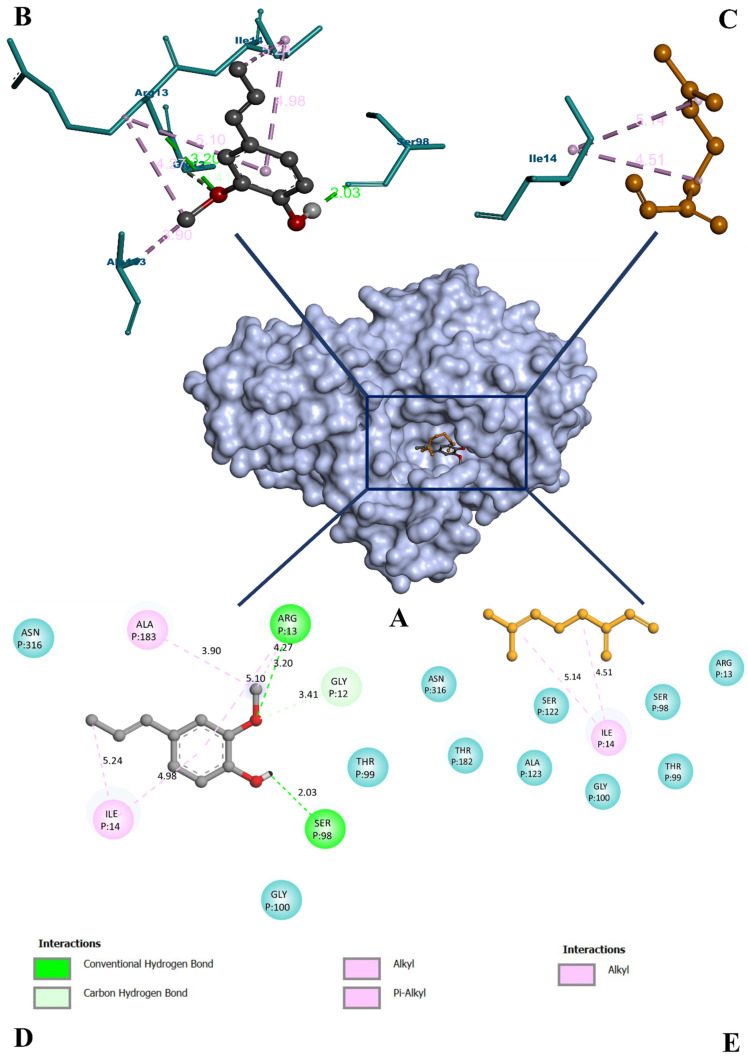
(**A**) The surface interaction view of GAPDH protein bound with isoeugenol (grey) and myrcene (orange). (**B**,**C**) The 3D representation of isoeugenol and myrcene binding to the residues, respectively. (**D**,**E**) The 2D representation of isoeugenol and myrcene binding to the residues, respectively. Teal: surrounding non-binding residues, colored: bound residues.

**Figure 8 molecules-27-02398-f008:**
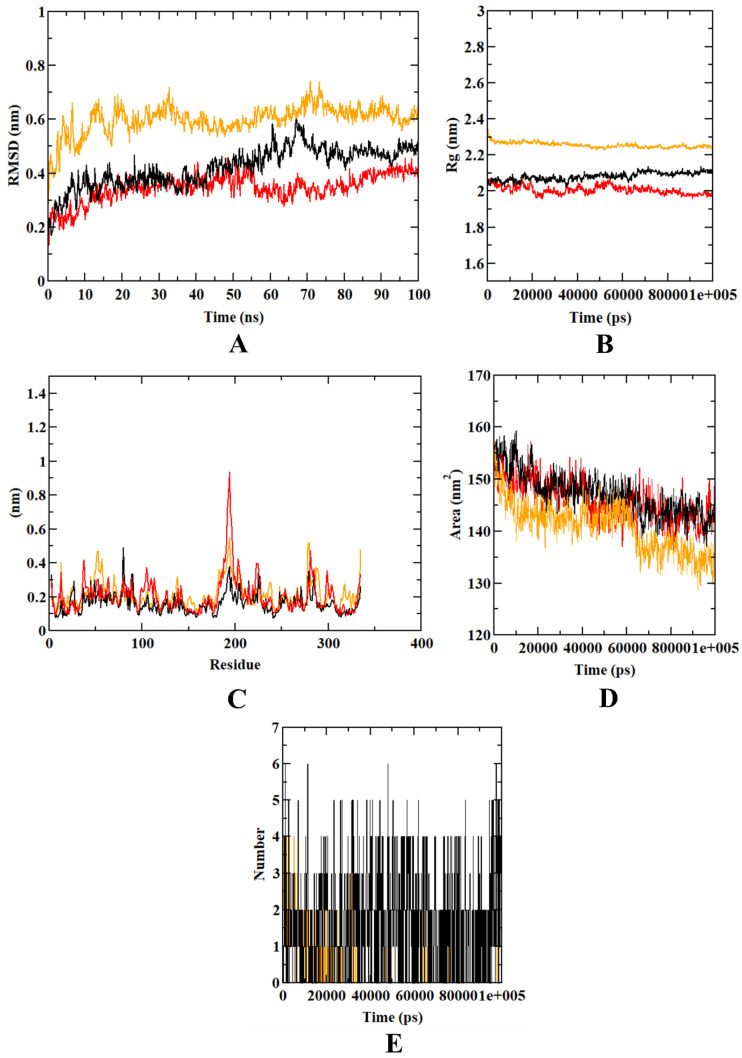
Analysis of RMSD, RMSF, Rg, SASA, and number of hydrogen bonds of GAPDH–isoeugenol complex (black), GAPDH–myrcene complex (orange), and GAPDH protein backbone atoms (red) at 100 ns. (**A**) RMSD of protein complex and protein backbone. (**B**) RMSF. (**C**) Radius of gyration (Rg). (**D**) SASA. (**E**) Ligand hydrogen bonds. Source like 1e+005 means 1 × 10^5^.

**Table 1 molecules-27-02398-t001:** Screening of active compounds using ADMET criteria.

Sl. No	Compound Names	Oral Bioavailability (OB ≥ 30%)	Blood–Brain Barrier (BBB)	Drug Half-Life (HL < 3 h)	Lipinski’s Rule (LR) of Five	Intestinal Epithelial Permeability (Caco-2 Cells)	Drug-Induced Liver Injury (DILI)	Clearness (CL > 15 mL/min/kg)	Molecular Weight (MW 100~600)	Hydrogen Bond Acceptor (0~12)	Hydrogen Bond Donor (0~7)	TPSA (0~140)	PAINS
1	Cyclo (L-val-L-Leu)	Pass	Pass	0.671	Accepted	−4.711	Negative	5.356	212.150	4	2	58.200	0
2	α-Cubebene	Fail	Pass	0.052	Accepted	−4.408	Negative	18.693	204.190	0	0	0.000	0
3	β-Caryophyllone	Moderate	Pass	0.227	Accepted	−4.750	Negative	14.774	220.180	1	0	17.070	0
4	Phytosterols	Pass	Pass	0.013	Accepted	−4.756	Negative	16.686	414.390	1	1	20.230	0
5	UNII-0V56HXQ8N5	Moderate	Moderate	0.059	Accepted	−4.357	Negative	19.832	204.190	0	0	0.000	0
6	(−)-Alloaromadendrene	Pass	Pass	0.040	Accepted	−4.577	Negative	13.563	204.190	0	0	0.000	0
7	(−)-Camphene	Pass	Pass	0.077	Accepted	−4.463	Negative	9.346	136.130	0	0	0.000	0
8	(−)-cis-Carveol	Fail	Pass	0.378	Accepted	−4.328	Negative	12.624	152.120	1	1	20.230	0
9	(−)-Linalool	Pass	Pass	0.493	Accepted	−4.375	Negative	9.738	154.140	1	1	20.230	0
10	(+)-α-Phellandrene	Pass	Pass	0.617	Accepted	−4.383	Negative	12.660	136.130	0	0	0.000	0
11	(+)-δ-Cadinene	Moderate	Moderate	0.051	Accepted	−4.469	Positive	7.421	204.190	0	0	0.000	0
12	(+)-endo-β-Bergamotene	Pass	Pass	0.063	Accepted	−4.466	Negative	16.946	204.190	0	0	0.000	0
13	(1S,2R,4S)-(−)-Bornyl acetate	Pass	Pass	0.243	Accepted	−4.552	Moderate	6.063	196.150	2	0	26.300	0
14	(1S)-1,7,7-Trimethylbicyclo[2.2.1]heptan-2-one	Pass	Pass	0.701	Accepted	−4.582	Negative	13.808	152.120	1	0	17.070	0
15	(E)-β-Farnesene	Fail	Fail	0.156	Accepted	−4.537	Moderate	13.186	204.190	0	0	0.000	0
16	(E)-α-Bisabolene	Fail	Fail	0.092	Accepted	−4.502	Negative	17.581	204.190	0	0	0.000	0
17	(E)-β-Ocimene	Pass	Pass	0.678	Accepted	−4.434	Negative	14.171	136.130	0	0	0.000	0
18	1-Octen-3-ol	Fail	Fail	0.672	Accepted	−4.256	Negative	7.650	128.120	1	1	20.230	0
19	1S-α-Pinene	Pass	Pass	0.114	Accepted	−4.303	Negative	15.022	136.130	0	0	0.000	0
20	2,3-Dimethylaniline	Pass	Pass	0.583	Accepted	−4.255	Negative	10.496	121.090	1	2	26.020	0
21	2,5-Dimethoxybenzoic acid	Pass	Moderate	0.885	Accepted	−4.853	Positive	7.488	182.060	4	1	55.760	0
22	3-Carene	Pass	Pass	0.132	Accepted	−4.307	Negative	16.061	136.130	0	0	0.000	0
23	4-Terpineol	Pass	Pass	0.447	Accepted	−4.217	Negative	14.345	154.140	1	1	20.230	0
24	Acetic acid	Pass	Pass	0.791	Accepted	−5.218	Negative	1.609	60.020	2	1	37.300	0
25	Acetyleugenol	Pass	Pass	0.843	Accepted	−4.453	Moderate	8.457	206.090	3	0	35.530	0
26	α-Fenchene	Pass	Pass	0.099	Accepted	−4.460	Negative	10.559	136.130	0	0	0.000	0
27	α-Humulene	Fail	No	0.095	Accepted	−4.425	Negative	8.432	204.190	0	0	0.000	0
28	α-Terpineol	Pass	Pass	0.527	Accepted	−4.193	Negative	8.942	154.140	1	1	20.230	0
29	Apigenin	Fail	No	0.856	Accepted	−4.847	Positive	7.022	270.050	5	3	90.900	0
30	Apigenin 7-glucuronide	Fail	No	0.715	Rejected	−6.376	Positive	1.194	446.080	11	6	187.120	0
31	β-Cadinene	Fail	Moderate	0.060	Accepted	−4.392	Negative	17.975	204.190	0	0	0.000	0
32	β-Carotene	Moderate	No	0.076	Rejected	−6.003	Negative	0.229	536.440	0	0	0.000	0
33	β-Caryophyllene	Pass	Pass	0.048	Accepted	−4.517	Negative	9.943	204.190	0	0	0.000	0
34	β-Pinene	Pass	Pass	0.107	Accepted	−4.460	Negative	10.097	136.130	0	0	0.000	0
35	Bis-acetic acid	Fail	No	0.998	Rejected	−7.722	Positive	12.869	1700.170	46	25	777.980	1 alert
36	Carotene	Pass	No	0.036	Rejected	−5.634	Negative	0.671	536.440	0	0	0.000	0
37	Carvacrol	Fail	Pass	0.671	Accepted	−4.436	Negative	11.335	150.100	1	1	20.230	1 alert
38	cis-Anethole	Pass	Moderate	0.638	Accepted	−4.440	Negative	11.146	148.090	1	0	9.230	0
39	Decanal	Fail	Pass	0.456	Accepted	−4.551	Negative	5.049	156.150	1	0	17.070	0
40	Dehydro-p-cymene	Pass	Pass	0.568	Accepted	−4.344	Moderate	10.755	132.090	0	0	0.000	0
41	Dipentene	Fail	Pass	0.233	Accepted	−4.320	Negative	11.517	136.130	0	0	0.000	0
42	Estragole	Moderate	Moderate	0.577	Accepted	−4.308	Negative	12.054	148.090	1	0	9.230	0
43	Eucalyptol	Pass	Pass	0.352	Accepted	−4.414	Negative	8.066	154.140	1	0	9.230	0
44	Eugenol	Fail	Pass	0.887	Accepted	−4.373	Negative	14.042	164.080	2	1	29.460	0
45	γ-Selinene	Pass	Pass	0.088	Accepted	−4.577	Negative	13.350	204.190	0	0	0.000	0
46	Geranyl acetate	Pass	Pass	0.506	Accepted	−4.420	Moderate	9.707	196.150	2	0	26.300	0
47	Isoeugenol	Pass	Moderate	0.880	Accepted	−4.579	Negative	13.435	164.080	2	1	29.460	0
48	L-Ascorbic acid	Fail	Fail	0.928	Accepted	−5.917	Positive	9.964	176.030	6	5	114.290	0
49	Linolenic acid	Fail	Fail	0.710	Accepted	−4.631	Negative	4.877	278.220	2	1	37.300	0
50	Luteolin-7-O-glucuronide	Fail	Fail	0.855	Rejected	−6.471	Positive	1.614	462.080	12	7	207.350	1 alert
51	Methyleugenol	Moderate	Pass	0.848	Accepted	−4.338	Negative	11.466	178.100	2	0	18.460	0
52	Molludistin	Fail	Fail	0.290	Accepted	−5.776	Positive	3.398	416.110	9	5	149.820	0
53	Myrcene	Pass	Pass	0.453	Accepted	−4.402	Moderate	13.108	136.130	0	0	0.000	0
54	Nerol	Fail	Pass	0.737	Accepted	−4.299	Positive	12.604	154.140	1	1	20.230	0
55	Octadeca-9,12-dienoic acid	Moderate	Fail	0.628	Accepted	−4.733	Negative	3.327	280.240	2	1	37.300	0
56	Octadecanoate	Fail	Fail	0.476	Accepted	−5.068	Negative	2.425	284.270	2	1	37.300	0
57	Oleic acid	Moderate	Fail	0.546	Accepted	−4.922	Negative	2.573	282.260	2	1	37.300	0
58	Orientin	Fail	Fail	0.724	Rejected	−6.208	Positive	5.042	448.100	11	8	201.280	1 alert
59	Palmitic acid	Fail	Fail	0.610	Accepted	−5.027	Negative	2.377	256.240	2	1	37.300	0
60	Thymol	Fail	Pass	0.682	Accepted	−4.387	Negative	9.444	150.100	1	1	20.230	0
61	Ursolic acid	Moderate	Pass	0.017	Accepted	−5.221	Negative	3.671	456.360	3	2	57.530	0

**Table 2 molecules-27-02398-t002:** Results from CytoNCA analysis revealing most significant target genes (selected on the basis of degree centrality).

Name	Betweenness Centrality	Closeness Centrality	Degree Centrality	Number of Undirected Edges
GAPDH	0.046608989	0.625061425	535	535
AKT1	0.038310118	0.625368732	535	535
ACTB	0.035911388	0.625061425	534	534
ALB	0.036238133	0.620487805	524	524
INS	0.047353182	0.619581101	515	515
TNF	0.026923424	0.610951009	497	497
IL6	0.025073263	0.610657705	491	491
TP53	0.028814967	0.591627907	428	428
IL1B	0.012693258	0.581618656	405	405
VEGFA	0.00953204	0.574525745	380	380
EGFR	0.018377788	0.576086957	374	374
STAT3	0.010150212	0.569127517	364	364
CTNNB1	0.01889698	0.56735058	350	350
MAPK3	0.010595927	0.563081009	333	333

**Table 3 molecules-27-02398-t003:** Binding affinity and non-bonding interactions of compounds with the target proteins.

Sl. No.	Name of the Compound	Binding Affinity (kcal/mol)	Total No. of Non-Bonding Interactions	Total No. of Hydrogen Bonds
1	1S-α-Pinene	−4.3	5	0
2	3-Carene	−4.3	5	0
3	4-Terpineol	−4.4	5	1
4	Acetyleugenol	−5.0	6	1
5	α-Terpineol	−4.4	5	1
6	β-Caryophyllene	−5.6	3	0
7	cis-Anethole	−4.1	4	1
8	Eucalyptol	−4.3	4	1
9	Geranyl acetate	−5.9	6	1
10	Isoeugenol	−6.0	7	2
11	Myrcene	−3.6	2	0

**Table 4 molecules-27-02398-t004:** Binding free energy calculations of protein–ligand complexes.

Types of Binding Free Energy	GAPDH–Isoeugenol Complex		GAPDH–Myrcene Complex	
	Values (kj/mol)	Standard Deviation (kj/mol)	Values (kj/mol)	Standard Deviation (kj/mol)
Van der Waal energy	−159.277	±7.426	−124.482	±11.201
Electrostatic energy	0.803	±1.153	1.251	±3.091
Polar solvation energy	26.262	±4.667	20.381	±6.330
SASA energy	−11.247	±0.590	−9.201	±2.998
Binding energy	−143.458	±8.685	−110.921	±10.120

## Data Availability

Not applicable.

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
