# Peer review of "Defining the Role of Isoeugenol from Ocimum tenuiflorum against Diabetes Mellitus-Linked Alzheimer’s Disease through Network Pharmacology and Computational Methods"

_molecules, 2022, doi:10.3390/molecules27082398_

Round 1

Reviewer 1 Report

In this paper, several techniques have been employed to study the link between diabetes mellitus and Alzhemier's disease. 
Overall, the paper is well done, but I have some issues:

  1. the introduction from line 92 to 110: I suggest to summarize only the main properties of the tulsi because it is too long. It should be better to give references where all these benefits are shown
  2. lines 188 and 199: please add references also for the method itself and not only for the papers where they are employed, such as 10.1517/17460441.2015.1032936 for MMPBSA
  3. figure 3, 4, 5 and 6 have a poor resolution
  4. figure 7A, please add some label to let the reader understand how the protein is rotated, such as the N-terminal and so on...
  5. why authors studied with MD only one compound? is it a matter of computational time? It should be better to have at least data for two molecules (the best two from docking or the best one and another one with a lower binding affinity) to have a comparison, otherwise it is difficult to conclude that isoeugenol is a good inhibitor. Without a comparison, every molecule that bind to the protein can be thought to be an inhibitor

Author Response

Response to reviewer comments:

All the authors thank the reviewer for their thoughtful suggestions to improve the quality of

the manuscript. The response to all the comments point-by-point are provided below.

Reviewer 1 comments

  1. The introduction from line 92 to 110: I suggest to summarize only the main properties of the tulsi because it is too long. It should be better to give references where all these benefits are shown.

Response: As per the reviewver’s suggestions, information about the tulsi plant has been summarized to give a consized yet meaningful introduction. Also, suitable references have been given wherever necessary.

  1. Lines 188 and 199: please add references also for the method itself and not only for the papers where they are employed, such as 10.1517/17460441.2015.1032936 for MMPBSA

Response: As per the reviewver’s suggestions, suitable references have been added for respective methods, including the one suggested by the reviewer.

  1. Figure 3, 4, 5 and 6 have a poor resolution.

Response: As per the reviewver’s suggestions, the authors have tried their best to enhance the image quality. The resolution of these images has been increased upto 1500 dpi using the software InkScape. Since open access softwares like CluePedia, CytoCluster, and, GO, Cytoscape does not provide high quality images with good resolution, it often becomes difficult for researchers to provide good quality images.

  1. Figure 7A, please add some label to let the reader understand how the protein is rotated, such as the N-terminal and so on...

Response: As per the reviewver’s suggestions, the protein molecule has been labelled properly using the literature avaialble in the RCSB PDB database. The protein molecule has been labelled with its N- and C-terminals, its rotation and position in the XYZ plane. The image has been added to supplemntary material (Supplementary Figure S1A).

  1. Why authors studied with MD only one compound? is it a matter of computational time? It should be better to have at least data for two molecules (the best two from docking or the best one and another one with a lower binding affinity) to have a comparison, otherwise it is difficult to conclude that isoeugenol is a good inhibitor. Without a comparison, every molecule that bind to the protein can be thought to be an inhibitor.

Response: As per the reviewer’s suggestions, a molecule with lowest binding affinity (myrcene) was selected as a negative control. Along with the MD simulation, the same compound has been used as a negative control even in molecular docking and MMPBSA experiments.

Reviewer 2 Report

The main concern with the manuscript is its contribution to knowledge. It is expected that a critical gap analysis will be done in the introductory section to justify the need to do this research. It is partially done by the authors, but must consider all aspects of the problem, including assumptions, limitations and restrictions, pros and cons, and relative merits of other publicly available studies and proposals.
- The intro paragraphs are too long. Authors should reorganize the introduction to form paragraphs of 5 to 6 lines. This makes reading more concise and organized for the MS reader.
- The protocol for molecular docking should be described in section 2.5
- Authors must create a redocking section to validate the docking protocol used. In this section the authors must perform the redocking of the crystallographic ligand, then the RMSD between the redocked ligand and the crystallographic ligand must be evaluated, if the RMSD was equal to or less than 2 angstroms we can say that the docking protocol is suitable for the system under study. According to the literature, this RMSD value is adequate
- The software used for md simulations must be mentioned. The protocol applied by the software must also be described.
- A separate section for affinity energy calculations must be created and its protocol must be described.

Author Response

Response to reviewer comments:

All the authors thank the reviewer for their thoughtful suggestions to improve the quality of

the manuscript. The response to all the comments point-by-point are provided below.

Reviewer 2 comments

The main concern with the manuscript is its contribution to knowledge. It is expected that a critical gap analysis will be done in the introductory section to justify the need to do this research. It is partially done by the authors, but must consider all aspects of the problem, including assumptions, limitations and restrictions, pros and cons, and relative merits of other publicly available studies and proposals.

Response: As per the reviewer’s suggestions, the entire introduction section has been re-written to be more precise and justified the need to do this resaerch by including more references.

  1. The intro paragraphs are too long. Authors should reorganize the introduction to form paragraphs of 5 to 6 lines. This makes reading more concise and organized for the MS reader.

Response: As per the reviewer’s suggestions, the introduction has been summarized to bring up only the essential information.

  1. The protocol for molecular docking should be described in section 2.5.

Response: As per the reviewer’s suggestions, protocol for molecular docking has been given in “Materials and Methods” section.

  1. Authors must create a redocking section to validate the docking protocol used. In this section the authors must perform the redocking of the crystallographic ligand, then the RMSD between the redocked ligand and the crystallographic ligand must be evaluated, if the RMSD was equal to or less than 2 angstroms we can say that the docking protocol is suitable for the system under study. According to the literature, this RMSD value is adequate.

Response: As per the reviewer’s suggestions, the docking protocol has been validated by re-docking the co-crystallized NAD+ with the protein and calculating the RMSD between bound and re-docked NAD+. The RMSD was found to be 2.10 Å, and the data has been added in the Supplementary Material (Supplementary Figure S1B).

  1. The software used for md simulations must be mentioned. The protocol applied by the software must also be described.

Response: As per the reviewer’s suggestions, name of the software used for MD simulations has been mentioned in the “Materials and Methods” section (GROMACS), along with the protocol used.

  1. A separate section for affinity energy calculations must be created and its protocol must be described.

Response: As per the reviewer’s suggestions, separate sections have been created for binding free energy calculations. Also, the protocol used has also been described in the “Materials and Methods” section.

Reviewer 3 Report

This manuscript deals with the integrated network pharmacology and phytoinformatics based research of compounds isolated from Ocimum tenuiflorum against diabetes mellitus linked Alzheimer’s disease. Based on the network analysis, most significant gene glyceraldehyde 3-phosphate dehydrogenase (GAPDH) was considered for docking study. From this screening, a computational hit was selected (isoeugenol). The manuscript fit in to the scopes of the Journal. Therefore, this referee believes that it deserves to be published in Molecules, pending the following minor points:

  • Natural products are often cited as an exception to Lipinski's rules. Please include this statement in the manuscript along with the proper references
  • Toxicity of computational hit selected (isoeugenol) should be mentioned in the manuscript.
  • Authors must explain why discriminate geranyl acetate as another computational hit.
  • CYP450 metabolism of isoeugenol generates the corresponding epoxide which could act as covalent inhibitor of GAPDH and other enzymes. Please comment about this possibility and cite the following article in the proper section:

Galbiati A, Zana A, Conti P. Covalent inhibitors of GAPDH: From unspecific warheads to selective compounds. Eur J Med Chem. 2020; 207: 112740. doi: 10.1016/j.ejmech.2020.112740

  • In figure 7, a protein-ligand interaction fingerprint (PLIF) analysis should be integrated
  • The simulations must be validated with preliminary in vitro enzymatic assays

Author Response

Response to reviewer comments:

All the authors thank the reviewer for their thoughtful suggestions to improve the quality of

the manuscript. The response to all the comments point-by-point are provided below.

Reviewer 3 comments

  1. Natural products are often cited as an exception to Lipinski's rules. Please include this statement in the manuscript along with the proper references.

Response: As per the reviewer’s suggestions, the statement has been incuded in the “Discussion” section of the manuscript, along with proper references.

  1. Toxicity of computational hit selected (isoeugenol) should be mentioned in the manuscript.

Response: As per the reviewer’s suggestions, toxicity results obtained from ADMET evaluation have been discussed in “Discussion” section of the manuscript.

  1. Authors must explain why discriminate geranyl acetate as another computational hit.

Response: As per the reviewer’s suggestions, authors have described why geranyl acetate was not considered for further analysis after molecualr docking. The compound was found with comparativley lower binding affinity and interactions compared to isoeugenol. The same has been discussed in “Discussion” section of the manuscript.

  1. CYP450 metabolism of isoeugenol generates the corresponding epoxide which could act as covalent inhibitor of GAPDH and other enzymes. Please comment about this possibility and cite the following article in the proper section: Galbiati A, Zana A, Conti P. Covalent inhibitors of GAPDH: From unspecific warheads to selective compounds. Eur J Med Chem. 2020; 207: 112740. doi: 10.1016/j.ejmech.2020.112740

Response: As per the reviewer’s suggestions, suitable information about the inhibition of GAPDH by the corresponding epoxide of isoeugenol obtained from CYP450 metabolism, and the same article has been cited, as suggested by the reviewer.

  1. In figure 7, a protein-ligand interaction fingerprint (PLIF) analysis should be integrated.

Response: As per the reviewer’s suggestions, PLIF analysis has been done for both isoeugenol and myrcene. The binding residues interacted with NAD+ have been used as reference for the analysis. The results have been given in the Supplementary Material (Supplementary Figure S2).

  1. The simulations must be validated with preliminary in vitro enzymatic assays.

Response: As per the reviewer’s suggestions, authors conducted a literature survey. But till date no phytochemicals have been used to inhibit the GAPDH. This is the first ever computational study that predicts isoeugenol from O. sanctum as a potential inhibitor. However, near in the future authors are planning for in vitro enzymatic inhibition of GAPDH using phytochemicals.

Round 2

Reviewer 1 Report

Authors replied to all my concerns and modified the manuscript accordingly. 

Reviewer 2 Report

The MS can be accepted for publication.